# Effect of Biochar Application on Morpho-Physiological Traits, Yield, and Water Use Efficiency of Tomato Crop under Water Quality and Drought Stress

**DOI:** 10.3390/plants12122355

**Published:** 2023-06-17

**Authors:** Abdullah Obadi, Abdulaziz Alharbi, Abdulrasoul Alomran, Abdulaziz G. Alghamdi, Ibrahim Louki, Arafat Alkhasha

**Affiliations:** 1Plant Production Department, King Saud University, Riyadh 11451, Saudi Arabia; aobadi@ksu.edu.sa (A.O.); akharbi@ksu.edu.sa (A.A.); 2Soil Science Department, King Saud University, Riyadh 11451, Saudi Arabia; agghamdi@ksu.edu.sa (A.G.A.); ibrahim.loukimi@hotmail.com (I.L.); aalkhasha@ksu.edu.sa (A.A.)

**Keywords:** date Palm biochar, fruit, growth, plant, salinity, soil amendment, tomato, water, yield

## Abstract

The use of saline water under drought conditions is critical for sustainable agricultural development in arid regions. Biochar is used as a soil amendment to enhance soil properties such as water-holding capacity and the source of nutrition elements of plants. Therefore, the experiment was conducted to evaluate the effects of biochar application on the morpho-physiological traits and yield of tomatoes under combined salinity and drought stress in greenhouses. There were 16 treatments consist two water quality fresh and saline (0.9 and 2.3 dS m^−1^), three deficit irrigation levels (DI) 80, 60, and 40% addition 100% of Evapotranspiration (ETc), and biochar application by rate 5% (BC_5%_) (*w*/*w*) and untreated soil (BC_0%_). The results indicated that the salinity and water deficit negatively affected morphological, physiological, and yield traits. In contrast, the application of biochar improved all traits. The interaction between biochar and saline water leads to decreased vegetative growth indices, leaf gas exchange, the relative water content of leaves (LRWC), photosynthetic pigments, and yield, especially with the water supply deficit (60 and 40% ETc), where the yield decreased by 42.48% under the highest water deficit at 40% ETc compared to the control. The addition of biochar with freshwater led to a significantly increased vegetative growth, physiological traits, yield, water use efficiency (WUE), and less proline content under all various water treatments compared to untreated soil. In general, biochar combined with DI and freshwater could improve morpho-physiological attributes, sustain the growth of tomato plants, and increase productivity in arid and semi-arid regions.

## 1. Introduction

The tomato plant (*Solanum lycopersicum* L.) considers one of the most popular and consumed vegetables in the world. In Saudi Arabia, a high yield of tomatoes is important to meet the increasing food demand. Tomatoes are rich in minerals and antioxidants such as phenols, lycopene, and vitamin C (VC) [1]. Salinity and drought are the most factors prominent abiotic stressors limiting crop growth and productivity worldwide [2,3]. Saudi Arabia is considered to be one of the driest areas in the world, with 85% of its water resources consumed for agriculture, in addition to other factors affecting agricultural activity [4]. Most of the soil in Saudi Arabia is sandy and sandy loam soils, which have a low water holding capacity, a high infiltration rate, and a low clay content and therefore need careful management. One of the shifts that the Kingdom has witnessed is water conservation.

Irrigated agriculture uses more than two-thirds of fresh water, making it the largest consumer of freshwater [5]. Producing high-quality food for an increasing global population and using water efficiently to irrigate crops is a major challenge for agriculture at present [6,7]. Adaptation of modern strategy for water-saving considers the key to increasing water use efficiency without a decline in productivity [8]. When tomato plants are subjected to water stress, they tend to reduce their leaf area and photosynthesis rate, which ultimately leads to reduced biomass accumulation and yield. Farooq, et al. [9] reported that drought stress can cause yield losses of 13% to 94%, depending on the intensity and duration of the drought. The reduction of tomato yield by 16.4% with moderate water deficit (75% ETc) compared with full irrigation (100% ETc) was noticed by [10]. However, drought stress often reduces yield and increases water use efficiency (WUE) as presented by [11]. 

Growing crops with saline soil or irrigating by saline water becomes a necessary measure to meet the increased food demand as a result of population increase, especially in areas where water supplies are often limited [12]. Soil salinization is one of the most harmful abiotic stresses to many cultivated crops around the world. Affect more than 20% of the irrigated land in the world, which slows plant growth and, as a result, lowers agricultural production [13]. The number of salt-affected regions increases mainly due to various natural and human factors, such as low rainfall, high temperatures, high evapotranspiration, and poor management and quality of irrigation water [14,15]. Soil salinity significantly decreases crop yield, particularly in vegetable crops. This related to the fact that vegetable crops usually have a low tolerance to salinity stress [16]. Ors, et al. [17] found that the interaction between salinity and drought led to a negative effect on all Morpho-physiological traits of tomato seedlings. Drought and salinity stresses leads to the generation of reactive oxygen species (ROS) in organelles such as chloroplasts, peroxisomes, and mitochondria. Moreover, ROS one of the major factors responsible for poor plant growth and productivity as a result of the peroxidation of cellular membrane lipids and degradation of enzyme proteins and nucleic acids [18].

Addition of biochar as an amendment was proposed as a method to improve long-term productivity and enhance water and fertilizer use efficiency. The international biochar initiative (IBI) defines biochar as a fine-grained organic material with a high carbon content that was produced through the pyrolysis process, which involves the thermal degradation of biomass at temperatures varied between 300 to 600 °C in the complete or partial absence of oxygen [19,20]. In recent years, the use of biochar in agricultural ecosystems obtained a lot of interest, the potential benefits of both yield and the environment for use the biochar [21,22]. Biochar and fertilizers may be the primary ways of enhancing soil fertility, water consumption efficiency, and crop yield in areas with limited water resources by reducing the negative consequences of drought stress [23]. In addition, the application of biochar enhances soil physical properties such as water holding capacity, structure, porosity, bulk density, and fertility [24,25]. Biochar increases soil water availability, resulting in reduced oxidative and osmotic stresses, thus improving plant growth and enhancing water uptake by plants [26]. The use of biochar indirectly improve soil water supply, altering soil structure and increasing water holding capacity [27]. Biochar has the potential to improve salt-affected sandy soil quality under arid conditions, thereby increasing vegetative growth and yield as well as the WUE of tomato plants [28]. The addition of biochar improved poor soil and increased vegetative growth traits, yield, and biomass of plants under salt and drought stress [29]. In another study, applying the biochar at rate of 4.8 t/ha led to an increase of the number tomato plant leaves, flowers, and fruit diameters, but this was not enough to make up for the reduction in fruit yield and increase levels of sodium ions that accumulated in the roots resulting from saline stress [30]. The main objective of using biochar relies on several factors, such as soil type, the amount of biochar added to the soil, and the physicochemical characteristics of biochar, which depend mainly on the type of feedstock and the pyrolysis conditions [28,31,32]. 

The majority of the studies were conducted under drought or salinity stress, with very few studies conducted under both drought and salinity stress having contrasting results mainly on the use of biochar. Therefore, the objectives of this study were to investigate the effect of salinity and drought stresses on the morpho-physiological, yield, and water use efficiency of tomato crops, as well as whether the use of date palm waste biochar produced could alleviate the negative effects of these stresses.

## 2. Results and Discussion

### 2.1. Morphological Traits of Tomato Plants

Salinity and water deficits have a negative effect on plant growth parameters, including plant height, leaf area, stem diameter, and wet and dry weight. On the other hand, the application of biochar improved all vegetative growth traits (Table 1). The stress of irrigation led to a significant decrease in most of the morphological characteristics, depending on the level and period of the stress [9]. Saline water impacted on plant vegetative growth attributes, due to a nutritional imbalance [33]. Moreover, a high salt concentration led to the inadequate development of the plant, due to osmotic stress and ion toxicity [34]. The addition of biochar increased the availability of the nutrition elements, which may enhance the morphological growth part of the plant [35]. Moreover, biochar increased the water availability in the soil, consequently reducing the impact of osmotic stress [26].

Plant height, leaf area index, stem diameter, and wet and dry weights were significantly affected by the interaction between salinity, drought stress, and biochar (Table 2). The addition of biochar positively affected the vegetative growth attributes in all irrigation treatments, especially when irrigated with freshwater. In contrast, the addition of biochar with saline water resulted in decreased vegetative growth traits, especially when the plants were subjected to water stress at 60% and 40% ETc (Table 2). The positive effects of biochar on vegetative growth traits is attributed to the stimulation of microbial activity in the root zone and the enhanced ability of the soil to retain water [36]. In addition, the biochar contains high amounts of minerals, such as calcium, magnesium, and inorganic carbon, which are beneficial for plant growth [37]. The biochar enhanced the soil water status and diluted the ion concentration under salinity stress, maintaining a suitable environment for plant growth [38]. Moreover, the addition of biochar improved vegetative growth due to the reduction in oxidative and osmotic stresses [39].

### 2.2. Physiological Parameters

Salinity and water deficits significantly decreased the leaf gas exchange traits (photosynthetic, conductivity, and transpiration rate) and LRWC, particularly with 60% and 40% ETc compared to 80% and 100% ETc. The salinity and the highest water deficit (S 2.3 ds m^−1^ and 40% ETc) increased the proline content in the leaves (Table 3). Many studies have shown that salinity and drought have adverse effects on plant growth, photosynthetic properties, and LRWC [40,41]. Ors et al. [17] found that increasing the salt concentration decreased gas exchange in the leaves of tomato seedlings. Our finding is in agreement with the finding of Alhoshan et al. [42] and Al-Harbi et al. [43], that the deficit irrigation significantly increased the proline content and the increase in proline percentage was associated with increased salinity and drought [44,45]. In contrast, the addition of biochar at a rate of 5% resulted in the highest leaf gas exchange traits, LRWC, and the lowest proline content in the leaves of tomatoes compared to untreated plants (Table 3). The increase in both gas exchange and LRWC, and the decrease proline content was due to the increasing water availability in the soil and salt leaching from the root zone. This reduces osmotic stress and enhances water uptake by the plant [26].

The addition of 5% biochar with freshwater led to the highest values for the leaf gas exchange traits under all water deficit treatments and an addition of 100% ETc compared to the untreated plants (without biochar), whereas the combination of salinity and deficit with 40% and 60% ETc negatively affected all the leaf gas exchange traits (Figure 1A–C). The results presented in Figure 1D illustrate that the highest proline content was recorded in the leaves of tomatoes grown under biochar with saline water at the highest water deficit of 40% ETc, while the lowest proline content was observed in the leaves irrigated with fresh water at 100% ETc. The highest LRWC values were obtained for all irrigation levels with biochar and freshwater, compared to the untreated plants (without biochar). In contrast, the lowest values for LRWC were found with biochar and irrigation with saline water under the highest water deficits of 40% and 60% ETc (Figure 1E). Alzahib et al. [46] found that increasing the salt concentration decreased the transpiration rate by 70.55%, the stomatal conductance by 7.13%, and the photosynthetic rate by 72.34% in the leaves of tomato seedlings. Based on the results reported by Akhtar et al. [47], the addition of biochar significantly increased the photosynthetic rate (Ph), the relative water content (RWC), and recorded the lowest proline content in tomato plants exposed to a water deficit. Similarly, Agbna et al. [48] observed that adding biochar to stressed and unstressed tomato plants significantly improved the photosynthetic and transpiration rates. Additionally, the use of biochar improved the leaf gas exchange and LWRC under salinity and drought stress conditions, indicating that biochar helped the plants retain firm leaves under abiotic stresses [27].

### 2.3. The Photosynthetic Pigments

Compared to plants that were not exposed to salinity and water deficit, the photosynthetic pigments traits (index of green leaves, chlorophyll a, chlorophyll b, total chlorophyll, and carotenoids) were reduced (Table 4). The decreased chlorophyll could be due to damage to the thylakoid membranes, as a result of the destructive effect of reactive oxygen species (ROS) on chloroplasts [49]. Salinity and water deficits caused a significant increase in the formation of ROS [50]. Another explanation for the decrease in chlorophyll content could be that the osmotic stress seriously damages the chloroplast layers by increasing the penetrability of the membrane [51]. For example, salt stress and drought have been shown to reduce the content of photosynthetic pigments in the leaves of tomatoes [26,50]. On the other hand, the addition of biochar resulted in an increase in the leaf green index, chlorophyll a, chlorophyll b, total chlorophyll, and carotenoids compared to the untreated plants (BC_0%_) (Table 4). Those results agreed with [52,53].

The highest values for the leaf pigments traits were recorded in plants treated with biochar and irrigated with fresh water under 100% of ETc compared to plants irrigated with saline water, particularly under the highest water deficit of 40% ETc, which recorded the lowest values (Table 5). Similar results were reported by Nadeem et al. [54], Kanwal et al. [55], and Karabay et al. [39], namely that the addition of biochar increased the chlorophyll content under salt stress and drought conditions. Additionally, Kul et al. [26] found that the application of 5% biochar improved the yield and growth characteristics of tomatoes grown under salinity conditions. Based on the results from our experiments, the use of biochar increases the photosynthesis rate, an indication of increased chlorophyll content.

### 2.4. Fruit Yield (kg m^−2^) and WUE (kg m^−3^) of Tomato Plants

Total tomato yield and WUE differ with the application of biochar, the water quality (saline and freshwater), and the irrigation deficit (Table 6). The results found that the addition of biochar increased the total yield and WUE. In contrast, irrigating with saline water resulted in a reduction in the total yield and WUE by 14.64% and 15.80%, respectively, compared to the control (full irrigation with 100% ETc). Similarly, deficit irrigation at 40% ETc resulted in a diminished total yield by 28.38% and an increased WUE by 79.01% compared to full irrigation at 100% ETc. Water and salt stress, as expected, have a detrimental impact on growth and yield, as confirmed by similar results stated by [16,56]. Some previous studies have shown that adding biochar can promote growth, increase yield, and improve WUE [48,57]. Guo et al. [58] found that adding 50 ton ha^−1^ of biochar increased the yield and WUE of tomatoes by 55.23% and 45.33%, respectively, compared to untreated plants.

The addition of 5% biochar with freshwater increased the yield of tomato plants under different irrigation treatments by 4.60%, 16.74%, 8.67%, and 2.97% for 100%, 80%, 60%, and 40% ETc, respectively, compared to the untreated plants (BC_0%_). The WUE increased by 97.02% for tomato plants, which were treated with biochar and irrigated with freshwater under a deficit irrigation of 40% ETc compared to full irrigation (Figure 2). The increase in yield and WUE with the biochar might be explained by its ability to retain water, improve porosity, and provide nutrients to the plant under water stress conditions. The increase in WUE with deficit irrigation could be attributed to reductions in the transpiration rate (TR) and stomatal closure in response to salt and water stress [48,59]. In contrast, the addition of biochar reduced the tomato yield by 42.48% when irrigated with saline water under the most severe stress conditions (40% ETc) compared to the control (Figure 2). It should be concluded that the negative effects from the biochar addition on the tomato yield in this study were most likely related to physiological drought resulting from the interaction between the biochar, saline water, and water deficit, and the high pH of biochar. As a result, the root absorption of water was more incomprehensible, leading to a decrease in the yield [60]. A high pH can affect the nutrient release into the soil, resulting in a decrease in the yield [61,62]. According to Hazman et al. [30], the addition of biochar to the soil improved some vegetative growth attributes, but did not mitigate the negative effects of salt stress on tomato fruit yield.

### 2.5. WUE Improvement and Irrigation Water Savings

The results in Table 6 indicate that saline water reduced the yield by 14.64% and the WUE by 15.80%. The results presented in Table 6 show that the irrigation deficit of 40% ETc reduced the tomato yield by 28.38%, while improving the WUE by 79.01% compared to the control (100% ETc). The addition of biochar at the rate (BC_5%_) specified increased the yield and WUE of tomato plants by 2.7% and 1.11%, respectively. This increase in the yield and WUE can be attributed to biochar behavior in the soil, promoting root growth in the soil. Similar results were reported by Obadi et al. [11] in the study on pepper plants grown in greenhouse, which indicated that the addition of biochar improved the WUE and irrigation water savings.

## 3. Materials and Methods

### 3.1. Experimental Site

The experiment was conducted in September 2021 to June 2022 under greenhouse conditions at Almohous Farms in the Thadiq region, 120 km northwest of Riyadh, Saudi Arabia. The average elevation was 722 m above sea level at latitude 25°17′40′′ N and longitude 45°52′55′′ E.

### 3.2. Treatments and Experiment Design

The experiment comprised of sixteen treatments combining two water quality treatments (0.9 and 2.3 dS m^−1^). The salinity of the irrigation water was prepared by adding sodium chloride (NaCl). Three deficit irrigation levels (80, 60, and 40%) based on crop evapotranspiration (ETc), in addition to full irrigation 100% of ETc as a control and practiced by farmers, and biochar application at a rate of 5% (*w*/*w*) (2.16 kg m^−2^) (BC_5%_) and untreated soil (BC_0%_). Experiments were designed as a randomized complete block (Split-Split-Plot Design) with three replicates. Water quality was the main factor, irrigation levels were sub-factors under the main, and biochar sub- factors the main. The treatments were distributed as follows: [Number of experimental units = 2 irrigation water quality **×** 4 irrigation levels **×** 2 biochar **×** 3 replicates = 48 experimental units]. The experimental unit consists of a line 6 m length and 1 m width, with emitters spaced 0.4 m (15 plants) and 1 m between the experimental units. the control was full irrigation (100% of ETc) without salinity and biochar (Figure 3).

The commercial tomato (Tone Guitar, a hybrid tomato) used for this study, was carried out in the greenhouse. The tomato (*Solanum lycopersicum* L.) seeds were planted in foam pots filled with peat moss: vermiculite (1:1 *v*/*v*) medium on 19 September 2021. Under controlled conditions in a fiberglass greenhouse, and regular practices for seedling growth at a temperature of 25 ± 2 °C in the daytime and 20 ± 2 °C at nighttime (to protect seedlings from the cold). Four weeks after sowing, seedlings were transferred to a uniform size with five leaves to the control greenhouse. The temperature and relative humidity (RH) in the control greenhouse were kept at 26 ± 1°C in the daytime, 19 ± 1 °C at night, and 75 ± 2% RH. Agricultural practices generally recommended for commercial tomato production under greenhouse conditions were employed, including soil sterilization, pest control, and fertilization. Fertilizers were applied by rate 285 kg N, 142 kg P, and 238 kg K per hectare as recommended by the local framers during the growth seasons. 

The surface drip irrigation system was designed inside the greenhouse. Based on the daily amount of evapotranspiration and crop coefficient (Kc) values, irrigation levels determined as calculated by [63] and were 40, 60, 80, and 100% of the crop water requirements (ETc). The ETc was calculated according to the following equation:

The surface drip irrigation system was designed inside the greenhouse. Based on the daily amount of evapotranspiration and crop coefficient (Kc) values, the irrigation levels determined as calculated by [63] were 40, 60, 80, and 100% of the crop water requirements (ETc). The ETc was calculated according to the following Equation:(1)ETc=Eo×Kp×Kc
where Eo is the evaporation from pan A (mm), Kp is the pan coefficient, and Kc is the crop coefficient.

### 3.3. Analysis of Water and Soil

Before the experiment, water and soil samples were collected from the greenhouse. A sample of sandy soil was air dried, passed through a 2 mm sieve, and a saturated soil paste extract was prepared. Analyzes of the water and soil samples, including the pH and EC, were performed using a pH (CG 817) and an EC (Test Kit Model 1500-20, Cole and Parmer) meter. Water-soluble sodium (Na^+^), magnesium (Mg^2+^), potassium (K^+^), calcium (Ca^2+^), and chloride (Cl^−^) were measured using an ion chromatography device (ICS-5000, Thermo Fisher Scientific, Waltham, MA, USA). Bicarbonate (HCO_3_^−^) and soluble carbonate (CO_3_^2−^) were measured using a titration method [64]. Chemical analysis of the water and soil is presented in Table 7.

### 3.4. Biochar Production

The biochar used in this experiment was prepared from date palm fronds waste at Al-Mohous Farms, 120 km northwest of Riyadh city. Biochar was produced by collecting the date palm waste and drying it in sunlight, then the fronds were cut into small pieces (15–20 cm). The biochar pieces were packed into a kiln. The kiln consisted of a tightly covered stainless-steel cylindrical container to reduce air volume and provide almost oxygen-free conditioning. The kiln underwent pyrolysis at a temperature of 450 °C ± 50 °C. The biochar was crushed manually and ground by an electrical grinder, and then sieved through a 2 mm sieve before mixing with the greenhouse soil at designated rates (Figure 4). More details about the preparation of biochar from date palm are described by [65,66]. Micromeritics ASAP 2020 BET was utilized to determine the surface area. An aqueous extract 1:10 (*w*/*v*) from the biochar was used for determining the pH and EC, which were measured with a pH meter, and a conductivity meter, respectively. The carbon (C), nitrogen (N) and hydrogen (H) contents were determined using a CHN analyzer (Series II; Perkin Elmer, Waltham, MA, USA). The moisture content, mobile materials, fixed carbon, and ash for the biochar were determined according to the ASTM D1762-84 method [67]. The chemical and physical properties of the obtained biochar are shown in Table 7.

### 3.5. The Measurements

#### 3.5.1. Growth and Physiological Parameters

The plant growth parameters were measured, including the plant height, stem diameter, and leaf area using a leaf area meter (LI-COR, Model 3000A), and the fresh and dry weight of the plant (leaves and stems). The dry weight was determined by a digital weighing balance after drying at 70 °C until the dry weight remained constant using a forced-air oven. The leaf tissue was used for the LRWC determination, measured as follows: leaf discs were sampled to obtain the fresh weight, followed by flotation on deionized water for up to 4 h to obtain the turgid weight. The dry weight was determined by oven drying the leaves at about 85 °C until they reached a constant weight. The LRWC was calculated according to [68].
(2)$$LRWC=\frac{fresh\;weight-dry\;weight}{turgid\;weight-dry\;weight}\times 100$$

Three mature leaves from the upper canopy of the plant were selected from each experimental unit to measure plant photosynthesis rate (Pn), transpiration rate (Tr), and conductivity (Cond) using a portable photosynthesize (Li-Cor, Lincoln, NE, USA). Chlorophyll a (Chol-a), chlorophyll b (Chol-b), total chlorophyll, and carotenoids are determined spectrophotometrically (T 80 UV/Visible Spectrophotometer, PG Instruments Ltd., Lutterworth, UK) according to [69]. The (Chol-a), (Chol-b), total chlorophyll, and carotenoids were calculated according to the following equations
(3)$$Chol.a=[\left(12.7\times \text{O.D}\;663\right)-(2.69\times \text{O.D}\;645)]\times V/1000\times W$$
(4)Chol.b=[(22.9×O.D 645)−(4.68×O.D 663)]×V/100
(5)$$Total\;Chol=[\left(20.2\times \text{O.D}\;645+\left(8.02\times \text{O.D}\;663\right)\right]\times V/1000\times W$$
(6)$$Carotenoids=\left[\text{O.D}\;480+\left(0.114\times \text{O.D}\;663\right)\right]\times \left(0.638\times \text{O.D}\;645\right)$$

O.D.: the extract’s optical density at the shown wavelength. V: the extract’s volume (mL). W: the fresh weight of leaves (g) [70]. Clausen’s method was followed to estimate the proline content in leaves [71].

#### 3.5.2. Total Yield and WUE

The amount of total yield and the weight of each fruit were measured using a digital balance throughout the harvesting time (kg /m^−2^). The WUE was calculated as the ratio of the total fresh fruit yield (TFFY, kg) to the cumulative amount of water applied (CIW, m^−3^) to the tomato plants throughout the growing season, according to [72]:(7)$$\mathrm{WUE}\;(\mathrm{kg}/m^{-3})=\frac{TFFY}{CIW}$$

The yield reduction (YR%) and amount of water saved (%) were determined using Equations (8) and (9), respectively, according to [73]. The WUE improvement was calculated using Equation (10), according to [11]:(8)$$YR\;(\%)=\left[\frac{\left(yield\;of\;control-yield\;of\;treatment\right)}{yield\;of\;control}\right]\times 100$$
(9)$$water\;saving\;(\%)=\left[\frac{\left(WCC-WCT\right)}{WCC}\right]\times 100$$
where WCC is the water consumption of the control (m^−3^/m^−2^) and WCT is the water consumption of the treatment (m^−3^/ m^−2^).
(10)$$Improve\;WUE\left(\%\right)=\left[\frac{\left(WUE\;of\;treatment-WUE\;of\;control\right)}{WUE\;of\;control}\right]\times 100$$

### 3.6. Statistical Analysis

ANOVA was applied to statistically analyze the data using SAS software, and the revised least significant difference (LSD) test was performed at the 0.05 confidence level [74].

## 4. Conclusions

The successful production of tomato crops in arid and semi-arid regions, having sandy soils characterized by low agricultural production factors, requires the addition of some amendments that mitigate the negative effects of salinity and drought. In this study, the addition of 5% biochar enhanced the morphological, physiological characteristics and WUE of tomatoes grown in greenhouse conditions. The yields from tomato crops irrigated with freshwater under various water deficit treatments were increased, by 4.60%, 16.74%, 8.67%, and 2.97%, for 100%, 80%, 60%, and 40% ETc, respectively, compared to the untreated plants (BC _0%_). Furthermore, the addition of biochar with saline water, especially at lower water supplies (40% ETc), decreased the vegetative growth, physiological traits, photosynthetic pigments, WUE, and yield by 42.48%. The addition of biochar to sandy soil could be recommended as an effective strategy to improve the growth and production of tomato plants under salinity or drought conditions, without interaction between them.

## Figures and Tables

**Figure 1 plants-12-02355-f001:**
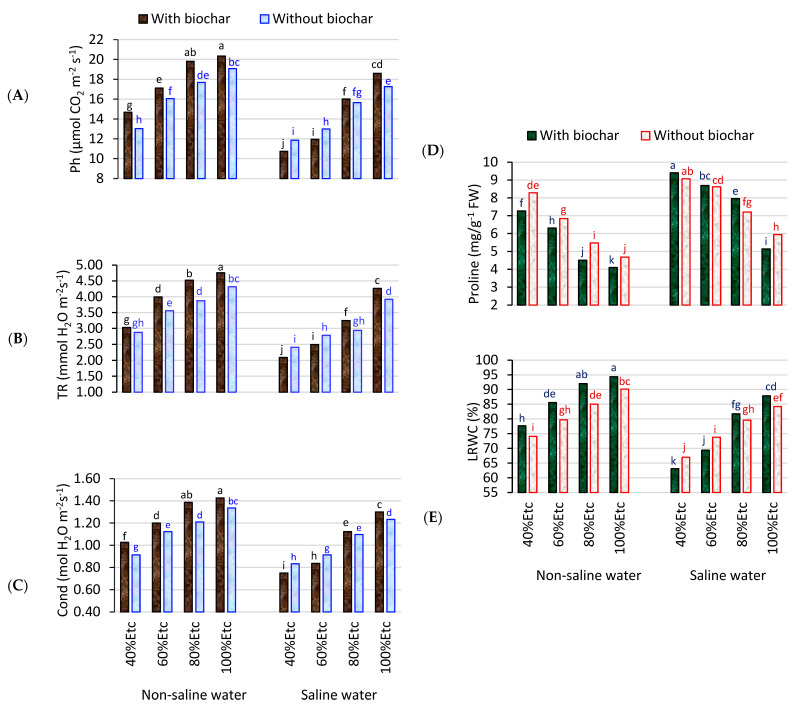
Interaction effects between salinity (S), deficit irrigation (DI), and biochar (BC) on the leaf photosynthetic rate (Ph) (**A**), the transpiration rate (TR) (**B**), the conductivity (Cond) (**C**), the proline (**D**), and the LRWC (**E**) of tomato leaves. Columns with the same letter are not significantly different at the 0.05 probability level, according to the LSD test.

**Figure 2 plants-12-02355-f002:**
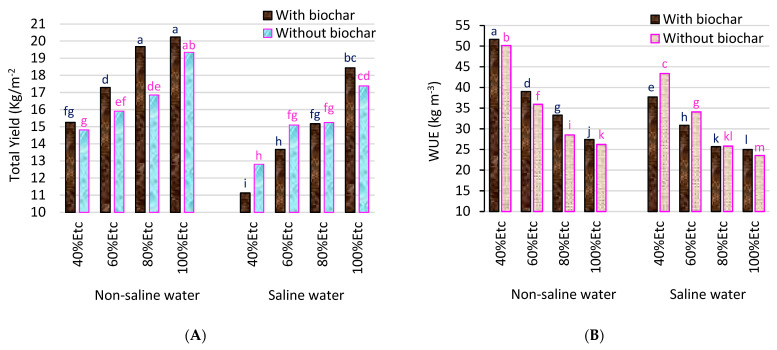
Interaction effects between salinity (S), water deficit (ETc), and biochar (BC) on total fruit yield (kg m^−2^) (**A**) and water use efficiency (WUE) (kg m^−3^) (**B**) for tomatoes. Columns with the same letter are not significantly different at the 0.05 probability level, according to the LSD test.

**Figure 3 plants-12-02355-f003:**
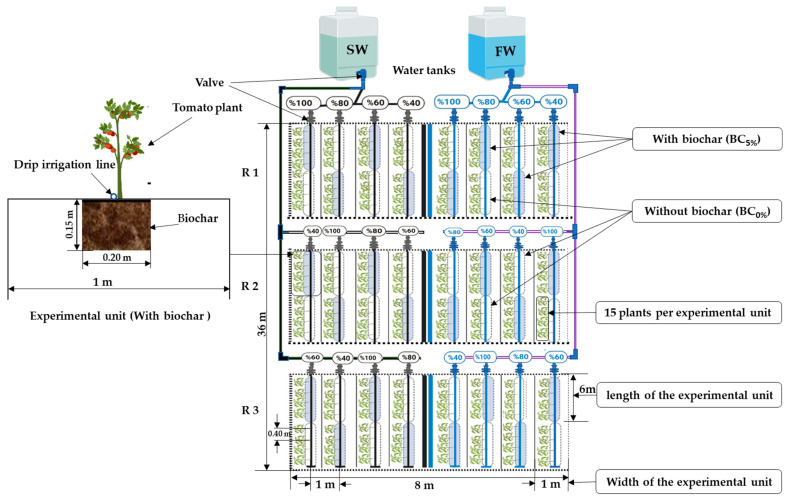
Sketch showing the experiment layout and randomization of the treatments.

**Figure 4 plants-12-02355-f004:**
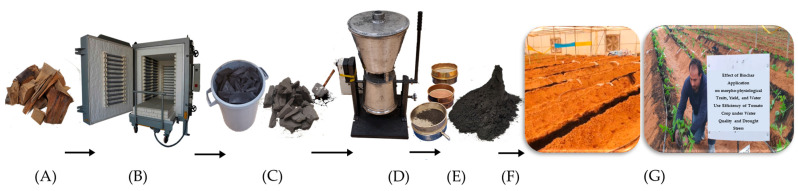
Schematic illustration of the biochar preparation: Date Palm (**A**) Kiln (**B**) Biochar (**C**,**F**) Electrical grinder (**D**) Sieve (**E**) Greenhouse (**G**).

**Table 1 plants-12-02355-t001:** The effects of salinity (S), biochar (BC), and irrigation water levels on tomato plant morphological traits such as plant height (cm), leaf area index (m^2^), stem diameter (mm), and fresh and dry weight (kg) and (g), respectively.

Treatments	Plant Height (cm)	Leaf Area Index (m^2^)	Stem Diameter (mm)	Fresh Weight of Plant (kg)	Dry Weight of Plant (g)
**Salinity**					
S _0.9 ds m^−1^_	334.05 a	0.73 a	15.65 a	1.77 a	223.43 a
S _2.3 ds m^−1^_	281.69 b	0.63 b	12.36 b	1.39 b	194.23 b
**Irrigation Levels (%ETc)**					
100	359.01 a	0.79 a	17.23 a	1.96 a	241.08 a
80	332.89 b	0.71 b	14.87 b	1.74 b	222.85 b
60	283.87 c	0.65 c	13.04 c	1.41 c	197.23 c
40	255.73 d	0.58 d	10.89 d	1.22 d	174.17 d
**Biochar**					
BC_0%_	303.60 b	0.67 b	13.58 b	1.54 b	203.08 b
BC_5%_	312.14 a	0.69 a	14.43 a	1.62 a	214.59 a

According to the LSD test, values with the same letter are not significantly different at the 0.05 probability level.

**Table 2 plants-12-02355-t002:** Interaction effects between salinity (S), deficit irrigation (DI) and biochar (BC) on tomato plant morphological traits such as plant height (cm), leaf area index (m^2^), stem diameter (mm), and fresh and dry weight (kg) and (g), respectively.

Salinity	Irrigation Levels (%ETc)	Biochar (%)	Height (cm)	Leaf Area Index (m^2^)	Stem Diameter (mm)	Fresh Weight of Plant (kg)	Dry Weight of Plant (g)
S _0.9 ds m^−1^_	100	BC_0%_	363.49 bc	0.81 b	18.14 b	1.99 b	234.75 c
		BC_5%_	383.82 a	0.85 a	19.17 a	2.14 a	255.00 ab
	80	BC_0%_	348.34 de	0.71 ef	15.49 de	1.82 d	221.15 de
		BC_5%_	366.63 b	0.79 bc	18.53 ab	2.07 ab	265.52 a
	60	BC_0%_	310.89 g	0.68 fg	13.85 fg	1.55 f	210.94 ef
		BC_5%_	334.07 f	0.76 cd	16.09 d	1.72 e	217.32 e
	40	BC_0%_	268.67 i	0.62 h	11.58 hi	1.40 h	181.76 h
		BC_5%_	296.51 h	0.66 g	12.38 h	1.47 gh	201.03 fg
S _2.3 ds m^−1^_	100	BC_0%_	336.18 ef	0.72 de	14.66 ef	1.79 de	231.48 cd
		BC_5%_	352.55 cd	0.76 c	16.94 c	1.91 c	243.10 bc
	80	BC_0%_	304.21 gh	0.67 fg	13.28 g	1.50 fg	195.82 g
		BC_5%_	312.35 g	0.66 g	12.18 h	1.56 f	208.91 ef
	60	BC_0%_	256.71 i	0.60 h	11.33 i	1.23 i	182.15 h
		BC_5%_	233.79 j	0.54 i	10.89 ij	1.13 j	178.49 hi
	40	BC_0%_	240.33 j	0.56 i	10.30 j	1.05 k	166.56 i
		BC_5%_	217.42 k	0.49 j	9.30 k	0.96 l	147.34 j

According to the LSD test, values with the same letter are not significantly different at the 0.05 probability level.

**Table 3 plants-12-02355-t003:** Effects of salinity (S), biochar (BC), and irrigation water levels on leaf gas exchange traits, proline content, and LRWC of tomato leaves.

Treatments	Photosynthesis Rate (µmol CO_2_ m^−2^ s^−1^)	Transpiration Rate (mmol H_2_O m^−2^s^−1^)	Conductivity (mol H_2_O m^−2^s^−1^)	Proline (mg/g^−1^ FW)	LRWC (%)
**Salinity**					
S _0.9 ds m^−1^_	17.23 a	3.87 a	1.20 a	5.94 b	84.84 a
S _2.3 ds m^−1^_	14.38 b	3.02 b	1.01 b	7.76 a	75.84 b
**Irrigation Levels (%ETc)**					
100	18.82 a	4.31 a	1.32 a	4.97 d	89.17 a
80	17.29 b	3.65 b	1.20 b	6.29 c	84.61 b
60	14.53 c	3.21 c	1.02 c	7.62 b	77.13 d
40	12.58 d	2.60 d	0.88 d	8.51 a	70.46 d
**Biochar**					
BC_0%_	15.45 b	3.34 b	1.08 b	7.02 a	79.20 b
BC_5%_	16.16 a	3.55 a	1.13 a	6.68 b	81.48 a

According to the LSD test, values with the same letter are not significantly different at the 0.05 probability level.

**Table 4 plants-12-02355-t004:** Effects of salinity (S), deficit irrigation (DI), and biochar (BC) on the leaf green index, chlorophyll a, chlorophyll b, total chlorophyll, and carotenoids in tomato plants.

Treatments	Leaf Green Index (SPAD)	Chlorophyll a (mg/g^−1^ FW)	Chlorophyll b (mg/g^−1^ FW)	Total Chlorophyll (mg/g^−1^ FW)	Carotenoids (mg/g^−1^ FW)
**Salinity**					
S _0.9 ds m^−1^_	48.63 a	2.55 a	1.11 a	3.66 a	4.91 a
S _2.3 ds m^−1^_	39.21 b	2.28 b	0.93 b	3.21 b	4.23 b
**Irrigation Levels (%ETc)**					
100	53.91 a	2.74 a	1.17 a	3.91 a	5.26 a
80	48.13 b	2.57 b	1.10 b	3.68 b	4.86 b
60	39.78 c	2.30 c	0.93 c	3.23 c	4.35 c
40	33.87 d	2.05 d	0.87 d	2.92 d	3.79 d
**Biochar**					
BC_0%_	42.72 b	2.36 b	1.00 b	3.36 b	4.47 b
BC_5%_	45.12 a	2.47 a	1.04 a	3.51 a	4.66 a

According to the LSD test, values with the same letter are not significantly different at the 0.05 probability level.

**Table 5 plants-12-02355-t005:** Interaction effects between salinity (S), deficit irrigation (DI), and biochar (BC) on the leaf green index, chlorophyll a, chlorophyll b, total chlorophyll, and carotenoids in tomato plants.

Salinity	Irrigation Levels (%ETc)	Biochar (%)	Leaf Green Index (SPAD)	Chlorophyll a (mg/g^−1^ FW)	Chlorophyll b (mg/g^−1^ FW)	Total Chlorophyll (mg/g^−1^ FW)	Carotenoids (mg/g^−1^ FW)
S _0.9 ds m^−1^_	100	BC_0%_	57.80 b	2.75 bc	1.12 cd	3.87 b	5.34 c
		BC_5%_	60.80 a	2.86 a	1.29 a	4.15 a	5.77 a
	80	BC_0%_	48.43 d	2.61 e	1.06 de	3.68 cd	4.71 fg
		BC_5%_	58.03 b	2.82 ab	1.26 ab	4.09 a	5.56 b
	60	BC_0%_	43.10 f	2.27 h	1.02 ef	3.28 ef	4.68 fg
		BC_5%_	45.53 e	2.63 de	1.14 c	3.77 bc	4.98 de
	40	BC_0%_	35.97 i	2.13 i	0.92 g	3.06 g	3.98 i
		BC_5%_	39.33 h	2.31 gh	1.06 de	3.37 e	4.25 h
S _2.3 ds m^−1^_	100	BC_0%_	46.37 e	2.64 de	1.06 de	3.71 cd	4.85 ef
		BC_5%_	50.67 c	2.69 cd	1.22 b	3.92 b	5.11 d
	80	BC_0%_	41.17 g	2.35 g	1.01 f	3.35 e	4.53 g
		BC_5%_	44.87 e	2.51 f	1.08 d	3.59 d	4.65 fg
	60	BC_0%_	36.97 i	2.24 h	0.91 g	3.15 fg	4.10 hi
		BC_5%_	33.50 j	2.05 j	0.67 h	2.72 h	3.66 j
	40	BC_0%_	31.93 j	1.90 k	0.89 g	2.79 h	3.60 j
		BC_5%_	28.23 k	1.87 k	0.61 i	2.48 i	3.32 k

According to the LSD test, values with the same letter are not significantly different at the 0.05 probability level.

**Table 6 plants-12-02355-t006:** The effects of salinity (S), deficit irrigation (DI), and biochar (BC) on a reduction in yield, saving water, total fruit yield (kg m^−2^), and WUE (kg m^−3^) (B) for tomato plants.

Treatments	Total Water Applied (m^−3^/ m^−2^)	Saving Water (%)	Total Yield (kg/ m^−2^)	Reduction in Yield (%)	WUE (kg m^−3^)	Improvement in WUE (%)
**Salinity**						
S _0.9 ds m^−1^_	------	------	17.42 a	00.00	36.53 a	00.00
S _2.3 ds m^−1^_	------	------	14.87 b	14.64	30.76 b	−15.80
**Irrigation Levels (%ETc)**						
100	0.738	0.00	18.85 a	0.00	25.54 d	00.00
80	0.591	19.92	16.74 b	11.19	28.34 c	10.96
60	0.443	39.98	15.50 c	17.77	34.98 b	36.96
40	0.295	60.03	13.50 d	28.38	45.72 a	79.01
**Biochar**						
BC_0%_	-------	------	15.93 b	0.00	33.46 b	0.00
BC_5%_	-------	------	16.36 a	−2.70	33.83 a	1.11

According to the LSD test, values with the same letter are not significantly different at the 0.05 probability level.

**Table 7 plants-12-02355-t007:** Physical–chemical properties of biochar, soil, and chemical properties of the water at the experimental location (greenhouse).

Parameters	Unit	Biochar	Soil		Fresh Water	Saline Water
Surface area	m^2^ g^−1^	237.80	---		---	---
pH	--	8.82	7.27		7.21	7.52
EC (dS m^−1^)	dS m^−1^	3.71	2.46		0.93	2.30
OM	%	30.33	Cations (meql^−1^)			
N	%	0.24	Ca^2+^	10.92	3.19	2.80
P	%	0.22	Mg^2+^	2.25	2.54	2.20
K	%	0.88	K^+^	5.10	0.13	0.29
C	%	60.0	Na^+^	3.80	4.70	21.04
H	%	3.44	Anions (meql^−1^)			
Ca	%	5.63	CO_3_^2−^	0.00	0.00	0.00
C/N ratio	-	250:1	Cl^−^	4.50	7.93	21.29
Moisture	%	3.53	HCO_3_^−^	18.3	2.32	2.86
Ash	%	25.70	SAR	2.02	2.78	13.32
Resident material	%	47.90		---	---	---
Physical properties of soil		Sand (%)	Silt (%)	Clay (%)	Soil texture	
		80	13	7	Loamy sand	

## Data Availability

Data available by request from the authors.

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
