# Peer review of "Effect of Biochar Application on Morpho-Physiological Traits, Yield, and Water Use Efficiency of Tomato Crop under Water Quality and Drought Stress"

_plants, 2023, doi:10.3390/plants12122355_

Round 1
Reviewer 1 Report
plants– Review Report
Effect of Biochar Application on Morpho-physiological traits,
Yield, and Water Use Efficiency of Tomato Crop under Water
Quality and Deficit
|
Paper Section |
Comment(s) |
|
Title |
Title is not so attractive. The physico-chemical attributes could have been better choice instead of Morpho-physiological traits. Drought stress instead of deficit could be better one. The first letter of connecting words or punctuations must be small. |
|
Abstract |
The Abstract of manuscript is not so attractive. the basic purpose of study is missing for example why there is need of selecting Biochar as world population increasing due to which there are serious threats to food security therefore need of organic amendments is increasing. These types of lines are necessary for background of study. The feed-stock type of which biochar was prepared is missing. The research gap is missing in abstract; it will make the abstract attractive if one or two lines should be added for research gape. |
|
Introduction |
The introduction is not according to the scientific write up style. Lahoz, et al. [10] reported the reduction in tomato yield is not attractive method in introduction portion. It will look more attractive if give citations at the end of sentences. The introduction is not well managed as all the information is summarized in two to three paragraphs. For each parameter and treatment there should be separate paragraph and should also be symmetrical. |
|
Results and discussion
|
Akhtar, et al. [37] reported that biochar can increase soil water and also citation is given at the end of sentence. One format should completely be followed. Discussion is not so briefed. Some more discussion is required with latest studies |
|
Material and Methods |
For pH and EC only, method is mentioned without proper citation. All other chemical and physical measuring methodologies are missing specially Ash content and Resident material. There is need to mention all the basic published methods for measuring these properties. . |
|
Conclusion |
The conclusion of the study is very short. Some more lines from results should be added for better and clear knowledge of study so that young researchers can easily review the manuscript for scientific write-up. Research gape and future recommendations are missing |
|
General Comment |
Overall, the manuscript is not so good.
The citation and references are not cross-referenced.
Manuscript is recommended for publication after revisions and thorough proof reading. Manuscript needs thorough proof reading and corrections before publication. |
Improve English language
Author Response
Reviewer No. 1
|
Paper Section |
Comment (s) |
|
Title |
Title is not so attractive. The physico-chemical attributes could have been better choice instead of Morpho-physiological traits. Drought stress instead of deficit could be better one. The first letter of connecting words or punctuations must be small.
Response: Thank you very much for your suggestion. The modification has been done |
|
Abstract |
The Abstract of the manuscript is not so attractive. the basic purpose of study is missing for example why there is need of selecting Biochar as world population increasing due to which there are serious threats to food security therefore need of organic amendments is increasing. These types of lines are necessary for background of study. The feed-stock type of which biochar was prepared is missing. The research gap is missing in abstract; it will make the abstract attractive if one or two lines should be added for research gape.
Response: We limited with number of words in the abstract, thus two sentences as an introduction is quite enough and then the objective …. |
|
Introduction |
The introduction is not according to the scientific write up style. Lahoz, et al. [10] reported the reduction in tomato yield is not attractive method in introduction portion. It will look more attractive if give citations at the end of sentences. The introduction is not well managed as all the information is summarized in two to three paragraphs. For each parameter and treatment there should be separate paragraph and should also be symmetrical.
Response: Some modification has been done in the introduction as labeled with red color. |
|
Results and discussion
|
Akhtar, et al. [37] reported that biochar can increase soil water and also citation is given at the end of sentence. One format should completely be followed. Discussion is not so briefed. Some more discussion is required with latest studies.
Response: This is a general comment. The modification has been done in a specific manner. |
|
Material and Methods |
For pH and EC only, method is mentioned without proper citation. All other chemical and physical measuring methodologies are missing specially Ash content and Resident material. There is need to mention all the basic published methods for measuring these properties.
Response: The methods, references, and devices used in all analyses of water, soil, and biochar were added. |
|
Conclusion |
The conclusion of the study is very short. Some more lines from results should be added for better and clear knowledge of study so that young researchers can easily review the manuscript for scientific write-up. Research gape and future recommendations are missing
Response: The modified and additions have been done. |
|
General Comment |
Overall, the manuscript is not so good. The citation and references are not cross-referenced. Manuscript is recommended for publication after revisions and thorough proof reading. The manuscript needs thorough proof reading and corrections before publication.
Response: The citation and reference style done according to journal forms all have been checked too. The manuscript went through a double check on Grammarly aspects and I think it's in good shape. |

Reviewer 2 Report
Dear Authors,
I reviewed your article titled in (Effect of Biochar Application on Morpho-physiological traits, Yield, and Water Use Efficiency of Tomato Crop under Water Quality and Deficit). Overall, the data presented here is valuable to those working in this field demonstrates the effectiveness of a relatively simple intervention that could be applied a wider scale especially in the field of improve production under abiotic stress. However, there are several previous works that study the same topic and the novelty is missed in this study. (https://doi.org/10.3390/agronomy13041039;10.1002/jsfa.12517;doi.org/10.3389/fmicb.2022.862075)
A thorough revision of the English grammar, sentence structure and deep editing of some parts needs to be undertaken before this is at all ready for publication. There are some other points that should be addressed in the individual sections, which I have specified below and in attached pdf file:
Title, Abstract, and introduction:
- Line 12-13: There are many previous works were handled with this issue. what is the novelty in your work?
Line 27: use words different than title and arrange alphabetically
Line 82-86: if you know that, why you did your work??
2. Results and discussion
Line 99: The discussion of the effect of drought is missed. Please revised.
Line 108-109: You show the same results twice? you have to present only the interaction if the parameter is significant (interaction is significant). If the interaction is not significant, then you present the individual effect.
Line 170: you have to present the interaction!!
Line 174: The whole MS needs English revised.
Line 191-192; 209, 211: This is poor discussion. You have to present the mode of action for your treatments
Line 214: no need for this table
3. Materials and Methods
Line 262: no need for these pictures since you mention the location
Line 269: You don’t mention any information regarding how you prepare the salinity water, how much you add per plant, how many times you add? Also, did you use any normal water?
Line 275: You dont mention any information regarding how you add the biochar, how much you add per plant, how many times you add? did you add at the surface or inside the growing media?
Line 288: Are these elements are added once? where are the micro elements? did you use hogland solution?
Line 300: It is better to add some pictures for this type of biochar or add chart shows the processes.
Line 321: did you measure the whole plant?
Some other comments are in the attached pdf file.

Need revision by native speaker
Author Response
Reviewer No. 2
Dear Authors,
I reviewed your article titled in (Effect of Biochar Application on Morpho-physiological traits, Yield, and Water Use Efficiency of Tomato Crop under Water Quality and Deficit). Overall, the data presented here is valuable to those working in this field demonstrates the effectiveness of a relatively simple intervention that could be applied a wider scale especially in the field of improve production under abiotic stress. However, there are several previous works that study the same topic and the novelty is missed in this study. (https://doi.org/10.3390/agronomy13041039;10.1002/jsfa.12517;doi.org/10.3389/fmicb.2022.862075).
Response: Thank you very much for your suggestion. The added to the references
A thorough revision of the English grammar, sentence structure and deep editing of some parts needs to be undertaken before this is at all ready for publication. There are some other points that should be addressed in the individual sections, which I have specified below and in attached pdf file:
Title, Abstract, and introduction:
- Line 12-13: There are many previous works were handled with this issue. what is the novelty in your work?
Response: Thank you very much for your suggestion. The previous work has been checked and added to the references.
Line 27: use words different than title and arrange alphabetically
Response: Keywords were modified accordingly.
Line 82-86: if you know that, why you did your work??
Response: Generally, research is done to confirm previous research or initiate a new one. In this study the use of biochar in the mitigation of abiotic stresses under different environments and soil than those in which previous studies were conducted.
- Results and discussion
Line 99: The discussion of the effect of drought is missed. Please revised.
Response: The discussion was revised as suggested by the reviewer.
Line 108-109: You show the same results twice? you have to present only the interaction if the parameter is significant (interaction is significant). If the interaction is not significant, then you present the individual effect.
Response: The interaction should be presented as its significance as well as the individual parameters.
Line 170: you have to present the interaction!!
Response: Presenting all parameters might be better to check the effect of biochar under different environments.
Line 174: The whole MS needs English revised.
Response: English was checked and many thanks to you for raising it.
Line 191-192; 209, 211: This is poor discussion. You have to present the mode of action for your treatments
Response: The discussion section was revised as suggested by the reviewer. Please check all the changes with the red color font.
Line 214: no need for this table
Response: Table 4 presents the individual effect of the parameters tested while Table 5 presents the interaction effect. As mentioned above both results are useful in order to check the individual effects of each parameter.
- Materials and Methods
Line 262: no need for these pictures since you mention the location
Response: Pictures removed.
Line 269: You don’t mention any information regarding how you prepare the salinity water, how much you add per plant, how many times you add? Also, did you use any normal water?
Response: Thank you very much for this important comment. The salinity of the irrigation water was increased by adding sodium chloride (NaCl) (saline water S 2.3 ds m-1). has been added
Line 275: You dont mention any information regarding how you add the biochar, how much you add per plant, how many times you add? did you add at the surface or inside the growing media?
Response: : Thank you very much for your suggestion. Before planting, biochar was put once at a line depth of 15 cm and 20 cm width, the length of the experimental unit, the application of biochar was 5% that is equal 2.16 kgm-2
Line 288: Are these elements are added once? where are the micro elements? did you use hogland solution?
Response: The micro and macronutrients were added in batches depending on the stages of the plant and depending on the recommended quantity to the tomato plant (usual farming).
Line 300: It is better to add some pictures for this type of biochar or add chart shows the processes.
Response: Thank you very much for your suggestion. Has been done.
Line 321: did you measure the whole plant?
Response: We selected some representative plant samples of the unit experiment
Some other comments are in the attached pdf file.

Reviewer 3 Report
Dear Authors,
I would like to thank you very much for the invitation as a reviewer for the manuscript Plants-2449559 “Effect of Biochar Application on Morpho-physiological traits, Yield, and Water Use Efficiency of Tomato Crop under Water Quality and Deficit” The article is interesting and dedicated to the vital problem of biochar application. The authors did a big job during this study, which should be highly appreciated. I hope that my remarks will help to improve some points of this article.
Line 16. This abbreviation (Etc) must be clarified.
Line 23. “… physiological …” what? This adjective is without noun.
Line 23. This abbreviation (WUE) must be clarified at the first mention.
Lines 92-97. It looks that it would be better to rephrase the aim of this study, because it is unclear what is the novelty of this research after other studies mentioned in the previous sentence.
Line 99. Here and further in the text. The subsections must also be numbered.
Lines 99-134. In this Subsection (Morphological Traits of Tomato Plants) there is a short literature review and tables with the results. Authors must add a table analyses and discussion of their OWN results!
Line 124. Authors have to explain how they present the significance of values in Tables: the letters should be considered for the whole column of separate factor. But it must be clarified in Materials and Method Section.
Line 175. What means “unirrigated”? According to the Materials and Methods there are 4 irrigation levels in the study.
Line 206, 232.. What treatment is used as a control? There is nothing mentioned in the Materials and Method Section.
Line 229. This abbreviation (TR) should be clarified.
Line 257. General remark to the Section: authors should add on the Materials and Methods description of both irrigation water qualities because it is important part of the experiment.
Line 267. The general note to Treatments and Experiment Design subsection: in the experiment design there is no Control or Standard treatments with usual practice in the region.
Line 270. It would be better to calculate the application rate in grams per square meter.
Lines 274-276. It would be better if authors will present the experiment design as a Table with clearly described treatments. How many plants was in one experimental unit?
Lines 288-289. It would be better to show the application rate per square meter or per plant.
Line 291. This abbreviation (Kc) should be clarified.
Line 295. It would be better to provide the explanation what is the “pan coefficient” and crop “coefficient” and how these coefficients are calculated.
Lines 310,321,323, 344. What is the brand of the device and country manufacturer? What was the accuracy of the weighing?
Line 316 (Table 7). Which methods and equipment were used to analyze soil on these parameters?
Line 357. This is three-factor experiment. It is necessary to explain how the data were processed - only for individual factors or the dispersion was calculated for a three-factor experiment.
Lines 370-372. There is no description about of the contribution of separate authors, Finding, Data Availability Statement.
Lines 400, 412, 482, 490, 495, 499, 521, 523, 524. The Latin species names must be in Italic.
Line 416, source 19. The bibliographical description of this source should be corrected according to Journal Rules.
Lines 503, 505, 513, 517, 518, 520, 522, 524. The name of the journal must be abbreviated.

Author Response
Reviewer No. 3
I would like to thank you very much for the invitation as a reviewer for the manuscript Plants-2449559 “Effect of Biochar Application on Morpho-physiological traits, Yield, and Water Use Efficiency of Tomato Crop under Water Quality and Deficit” The article is interesting and dedicated to the vital problem of biochar application. The authors did a big job during this study, which should be highly appreciated. I hope that my remarks will help to improve some points of this article.
Line 16. This abbreviation (Etc) must be clarified.
Response: All abbreviations in the manuscript has been clarified.
Line 23. “physiological” what? This adjective is without noun.
Response: clarified too with many thanks.
Line 23. This abbreviation (WUE) must be clarified at the first mention.
Response: Again, thanks you it has been corrected.
Lines 92-97. It looks that it would be better to rephrase the aim of this study, because it is unclear what is the novelty of this research after other studies mentioned in the previous sentence.
Response: The novelty of this work is done on different environments and soil types than those in which previous studies were conducted.
Line 99. Here and further in the text. The subsections must also be numbered.
Response: All subsections were numbered as suggested
Lines 99-134. In this Subsection (Morphological Traits of Tomato Plants) there is a short literature review and tables with the results. Authors must add a table analyses and discussion of their OWN results!
Response: The discussion section was revised and all changes in red color of the text.
Line 124. Authors have to explain how they present the significance of values in Tables: the letters should be considered for the whole column of separate factor. But it must be clarified in Materials and Method Section.
Response: A statement of statistical analysis was added to the text.
Line 175. What means “unirrigated”? According to the Materials and Methods there are 4 irrigation levels in the study.
Response: This word has been corrected and deleted. It was used under salinity conditions.
Line 206, 232 What treatment is used as a control? There is nothing mentioned in the Materials and Method Section.
Response: The control (full irrigation (100% of ETc without salinity and biochar). It has been added.
Line 229. This abbreviation (TR) should be clarified.
Response: Transpiration rate and was added
Line 257. General remark to the Section: authors should add on the Materials and Methods description of both irrigation water qualities because it is important part of the experiment.
Response: Thank you very much for your suggestion. The analyses of water and how to prepare saline water were added
Line 267. The general note to Treatments and Experiment Design subsection: in the experiment design there is no Control or Standard treatments with usual practice in the region.
Response: Again, full irrigation is the control treatment. We tried to introduce deficit irrigation to farmers in order to save water in addition to biochar
Line 270. It would be better to calculate the application rate in grams per square meter.
Response: The application of biochar was 5% which is equal to 2.16 kg m-2
Lines 274-276. It would be better if authors will present the experiment design as a Table with clearly described treatments. How many plants was in one experimental unit?
Response: The scheme of the experiment was added, with all the dimensions of each unit.
Lines 288-289. It would be better to show the application rate per square meter or per plant.
Response: The application rate of biochar was added and equal to 2.6 kg m-2
Line 291. This abbreviation (Kc) should be clarified.
Response: kc is crop coefficient and defined too.
Line 295. It would be better to provide the explanation what is the “pan coefficient” and crop “coefficient” and how these coefficients are calculated.
Response: This well-known coefficient uses pan evaporation to schedule irrigation. We think defining these coefficients is sufficient for the article.
Lines 310,321,323, 344. What is the brand of the device and country manufacturer? What was the accuracy of the weighing?
Response: A full paragraph was added on the analysis of soil, biochar, and water with a diagram of how to prepare the biochar too.
Line 316 (Table 7). Which methods and equipment were used to analyze soil on these parameters?
Response: same as above.
Line 357. This is three-factor experiment. It is necessary to explain how the data were processed - only for individual factors or the dispersion was calculated for a three-factor experiment.
Response: Experiments were designed as a randomized complete block (Split-Split-Plot Design) with three replicates. Water quality was the main factor, irrigation levels were factors under the main, and biochar factors sub-under the main and we used SAS Program, version 2019 to analyze the data
Lines 370-372. There is no description about of the contribution of separate authors, Finding, Data Availability Statement.
Response: The authors' contribution and conflict of interest were added through the submission.
Lines 400, 412, 482, 490, 495, 499, 521, 523, 524. The Latin species names must be in Italic.
Response: Done with many thanks to you.
Line 416, source 19. The bibliographical description of this source should be corrected according to Journal Rules.
Response: Done according to the journal forms and rules.
Lines 503, 505, 513, 517, 518, 520, 522, 524. The name of the journal must be abbreviated.
Response: Done with many thanks to you.

Round 2
Reviewer 2 Report
Dear authors,
Thank you for your improvement in your article. However, the most important comments were not replied or misunderstand replied.
1- I wrote: However, there are several previous works that study the same topic and the novelty is missed in this study. (https://doi.org/10.3390/agronomy13041039;10.1002/jsfa.12517;doi.org/10.3389/fmicb.2022.862075).
Your Response: Thank you very much for your suggestion. The added to the references
New Comment: this is not the reply to my comment. The novelty of your work should be addressed. As I said there are several same previous articles that handling with your subject.
2- I wrote: Line 12-13: There are many previous works were handled with this issue. what is the novelty in your work?
Your Response: Thank you very much for your suggestion. The previous work has been checked and added to the references.
New Comment: this is not the reply to my comment. The novelty of your work should be addressed. As I said there are several same previous articles that handling with your subject.
2-I wrote: Line 108-109: You show the same results twice? you have to present only the interaction if the parameter is significant (interaction is significant). If the interaction is not significant, then you present the individual effect.
Response: The interaction should be presented as its significance as well as the individual parameters.
New Comment: if the interaction is significant; you have to present the interaction only. If the interaction is not significant; you have to present, the individual parameters only. I hope you get my point of view.
3- Table one: Add the SE or SD to every value (for all tables). Also, the units of leaf area and fresh weight could be converts to m2 and kg, respectively.
4- Figure 2: remove the horizontal lines.
5- Figure 4G: there is other langue than English. Please remove it.

Author Response
Thank you for your improvement in your article. However, the most important comments were not replied or misunderstand replied.
- I wrote: However, there are several previous works that study the same topic and the novelty is missed in this study.
(https://doi.org/10.3390/agronomy13041039;10.1002/jsfa.12517;doi.org/10.3389/fmicb.2022.862075)
Your Response: Thank you very much for your suggestion. The added to the references
New Comment: this is not the reply to my comment. The novelty of your work should be addressed. As I said there are several same previous articles that handling with your subject.
Response: The novelty of this work is done on different environments and soil types. Generally, research is done to confirm previous research or initiate a new one. In this study the use of biochar in the mitigation of abiotic stresses under different environments and soil than those in which previous studies were conducted. The reference or the previous work you mentioned was added to the references. Again, as we mentioned, rephrase the objectives to include the conflict research on the use of biochar. The following are some of the previous works mentioned in the manuscripts.
Hazman, M.Y.; El-Sayed, M.E.; Kabil, F.F.; Helmy, N.A.; Almas, L.; McFarland, M.; Shams El Din, A.; Burian, S. Effect of biochar application to fertile soil on tomato crop production under Saline irrigation regime. Agronomy 2022, 12, 1596.
Alzahib, R.H.; Migdadi, H.M.; Al Ghamdi, A.A.; Alwahibi, M.S.; Ibrahim, A.A.; Al-Selwey, W.A. Assessment of morpho-physiological, biochemical and antioxidant responses of tomato landraces to salinity stress. Plants 2021, 10, 696.
- Akhtar, S.S.; Li, G.; Andersen, M.N.; Liu, F. Biochar enhances yield and quality of tomato under reduced irrigation. Agric. Water Manag. 2014, 138, 37-44.
- Agbna, G.H.; Dongli, S.; Zhipeng, L.; Elshaikh, N.A.; Guangcheng, S.; Timm, L.C. Effects of deficit irrigation and biochar addition on the growth, yield, and quality of tomato. Sci. Hortic. 2017, 222, 90-101.
Zhang, W.; Wei, J.; Guo, L.; Fang, H.; Liu, X.; Liang, K.; Niu, W.; Liu, F.; Siddique, K.H. Effects of two biochar types on mitigating drought and salt stress in tomato seedlings. Agronomy 2023, 13, 1039. (Suggested by the reviewers)
2- I wrote: Line 12-13: There are many previous works were handled with this issue. what is the novelty in your work?
Your Response: Thank you very much for your suggestion. The previous work has been checked and added to the references.. Please check the objectives and our answer above too.
New Comment: this is not the reply to my comment. The novelty of your work should be addressed. As I said there are several same previous articles that handling with your subject.
Response: The novelty of this work is done on different environments and soil types than those in which previous studies were conducted (same as above).
2-I wrote: Line 108-109: You show the same results twice? you have to present only the interaction if the parameter is significant (interaction is significant). If the interaction is not significant, then you present the individual effect.
Response: The interaction should be presented as its significance as well as the individual parameters.
New Comment: if the interaction is significant; you have to present the interaction only. If the interaction is not significant; you have to present, the individual parameters only. I hope you get my point of view.
Response: Yes indeed, I got your point but we think having both individual and interaction strengthens the work
3- Table one: Add the SE or SD to every value (for all tables). Also, the units of leaf area and fresh weight could be converts to m2 and kg, respectively.
Response: Thank you very much for your suggestion. The modification has been done on the units. Concerning SE and SD we thing the LSD cover both in statistical analysis.
4- Figure 2: remove the horizontal lines.
Response: The line has been removed.
5- Figure 4G: there is other langue than English. Please remove it.
Response: Has been removed.

Round 3
Reviewer 2 Report
Thank you for your improvement